# Usefulness of the Trabecular Bone Score in Assessing the Risk of Vertebral Fractures in Patients with Cirrhosis

**DOI:** 10.3390/jcm11061562

**Published:** 2022-03-12

**Authors:** Yui Ogiso, Tatsunori Hanai, Kayoko Nishimura, Takao Miwa, Toshihide Maeda, Kenji Imai, Atsushi Suetsugu, Koji Takai, Masahito Shimizu

**Affiliations:** 1Department of Gastroenterology/Internal Medicine, Gifu University Graduate School of Medicine, Gifu 501-1194, Japan; g104053@yahoo.co.jp (Y.O.); miwa_t@gifu-u.ac.jp (T.M.); m0a2t1o8@gifu-u.ac.jp (T.M.); ikenji@gifu-u.ac.jp (K.I.); asue@gifu-u.ac.jp (A.S.); koz@gifu-u.ac.jp (K.T.); shimim@gifu-u.ac.jp (M.S.); 2Center for Nutrition Support and Infection Control, Gifu University Hospital, Gifu 501-1194, Japan; km@gifu-u.ac.jp

**Keywords:** bone microarchitecture, bone mineral density, cirrhosis, osteoporosis, vertebral fracture

## Abstract

The trabecular bone score (TBS), a surrogate measure of bone microarchitecture, provides complementary information to bone mineral density (BMD) in the assessment of osteoporotic fracture risk. This cross-sectional study aimed to determine whether TBS can identify patients with liver cirrhosis that are at risk of vertebral fractures. We enrolled 275 patients who completed evaluations for lumbar BMD, TBS, and vertebral fractures between November 2018 and April 2021. BMD was measured using dual-energy X-ray absorptiometry (DXA), TBS was calculated by analyzing DXA images using TBS iNsight software, and vertebral fractures were evaluated using Genant’s semi-quantitative method with lateral X-ray images. Factors associated with vertebral fractures and their correlation with the TBS were identified using regression models. Of the enrolled patients, 128 (47%) were female, the mean age was 72 years, and 62 (23%) were diagnosed with vertebral fractures. The prevalence of vertebral fractures was higher in women than in men (33% vs. 14%; *p* < 0.001). The unadjusted odds ratio (OR) of the vertebral fractures for one standard deviation decrease in TBS and BMD was 2.14 (95% confidence interval [CI], 1.69–2.73) and 1.55 (95% CI, 1.26–1.90), respectively. After adjusting for age, sex, and BMD, the adjusted OR of the vertebral fractures in TBS was 2.26 (95% CI, 1.52–3.35). Multivariate linear regression analysis showed that TBS was independently correlated with age (β = −0.211), body mass index (β = −0.251), and BMD (β = 0.583). TBS can help identify patients with cirrhosis at risk of vertebral fractures.

## 1. Introduction

Osteoporosis is defined as a systemic skeletal disease characterized by the low bone mass and microarchitectural deterioration of bone tissue, with a consequent increase in bone fragility and susceptibility to fracture [1,2,3]. Fractures associated with osteoporosis generally increase after the age of 55 years in women and after 65 years in men, with profound health consequences for older people [3]. Therefore, osteoporosis has become a critical health issue in an aging society.

Osteoporosis is a common complication of cirrhosis, occurring in 20–30% of patients [4]. However, it is often overlooked in patients with cirrhosis. Patients with cirrhosis are approximately twice as likely to develop osteoporotic fractures as those without cirrhosis [5]. Vertebral fractures, the most frequent type of osteoporotic fracture, occur in 7–35% of patients with cirrhosis whereas the incidence of peripheral fractures is approximately 10%, indicating that bone loss in liver cirrhosis is more severe in trabecular bone, such as the vertebra, than in cortical bone [6]. Vertebral fractures are associated with an increased risk of incident vertebral fractures and other osteoporotic fractures, mortality, and ad-verse health outcomes [3]. Therefore, the assessment of bone strength and fracture risk is clinically important to identify patients with cirrhosis at high risk for vertebral fractures.

Bone mineral density (BMD) measured by dual-energy X-ray absorptiometry (DXA) is a key determinant of bone strength and fracture risk. Although BMD is widely used to diagnose osteoporosis and estimate osteoporotic fracture risk [7,8,9], most osteoporotic fractures occur in individuals with BMD values in the osteopenia or normal range [9,10]. This suggests that the BMD measurement is limited in estimating osteoporotic fracture risk and that other fracture risk factors should be considered for a more accurate assessment of bone strength and fracture risk.

Bone microarchitecture is another important determinant of bone strength and fracture risk [11], which can be directly assessed by bone biopsy or indirectly assessed by imaging technologies [12]. However, its routine evaluation is inadequate owing to invasive procedures, cost involved, and limited availability in facilities. Recently, the trabecular bone score (TBS), which measures the gray-level texture from previously obtained two-dimensional DXA images of the lumbar spine, has attracted much attention as a simple alternative to measure bone microarchitecture [11,12,13,14]. TBS is highly correlated, albeit indirectly, with three-dimensional parameters of bone microarchitecture such as the trabecular number, trabecular separation, connective density, and structure model index [11]. TBS predicts osteoporotic fractures in primary osteoporosis and some secondary osteoporosis, independent of BMD [15,16].

TBS measurement is important in assessing bone strength; however, it remains unclear whether TBS is useful in identifying patients with liver cirrhosis at risk for vertebral fractures. The purpose of this study was to characterize cirrhotic patients with and without vertebral fractures, to determine the predictive validity of TBS for vertebral fractures, and to identify risk factors associated with reduced TBS.

## 2. Materials and Methods

### 2.1. Study Design

Among the 299 cirrhotic patients treated at Gifu University Hospital (Gifu, Japan) between November 2018 and April 2021, 24 patients who were not evaluated for BMD, TBS, or vertebral fractures were excluded from the data analysis, and consequently, the remaining 275 were enrolled in this cross-sectional study. The study protocol was reviewed and approved by the Institutional Review Board of the Gifu University Graduate School of Medicine (approval number: 2021-B123). The study was performed in accordance with ethical standards laid down in the Declaration of Helsinki 1964 and its later amendments.

Liver cirrhosis was diagnosed by hepatologists at our hospital based on a combination of clinical features, such as ascites, esophagogastric varices, portosystemic shunts, laboratory variables, medical imaging features, and, if possible, histological features. The severity of liver disease was estimated using the Child–Pugh classification and the model for end-stage liver disease (MELD) score, and the diagnosis of hepatocellular carcinoma (HCC) was based on a combination of typical imaging characteristics or histological features [17].

Inclusion criteria were patients aged ≥20 years, those with cirrhosis of any etiology, and the assessment of BMD, TBS, and vertebral fractures. Exclusion criteria included those who refused to provide informed consent, had previous organ transplantation, non-hepatic malignancies, bone metastasis of any malignancy, pregnant women, those who use implantable medical devices, such as pacemakers or defibrillators, and those with unstable medical conditions (severe sepsis, heart failure, respiratory failure, renal failure, and other acute life-threatening diseases). Data on baseline characteristics within 1 month of the bone strength assessment obtained from our prospectively collected database were evaluated.

### 2.2. Measurement of Bone Strength

Lumbar spine BMD was measured using a DXA bone densitometer (Horizon W, Hologic Inc., Marlborough, MA, USA), and the values for the lumbar spine (L2–L4) were calculated using densitometric software (Hologic APEX software version 5.5.3.1, Hologic Inc., Marlborough, MA, USA). TBS was calculated by analyzing the raw data of each DXA image at the same region of BMD measurement using the TBS iNsight software (TBS iNsight version 3.0; Medimaps Group, Geneva, Switzerland). The strength of bone microarchitecture was classified into the three following grades according to TBS: TBS > 1.31, normal structure (low risk of fracture); TBS = 1.23–1.31, partially degraded structure (intermediate risk); and TBS < 1.23, degraded structure (high risk) [15]. Lumbar spine TBS, unlike BMD, was not influenced by osteoarthritis and pre-existing vertebral compression fractures [18].

### 2.3. Diagnosis of Vertebral Fractures

Vertebral fractures were assessed by Genant’s semiquantitative method using lateral X-ray images of the thoracic and lumbar spine [19]. This method classifies the degree of vertebral deformation into the four following grades by visual inspection of the lateral X-ray images without directly measuring the vertebra: Grade 0, normal; Grade 1, 20–25% decrease in anterior, middle, and posterior height and 10–20% decrease in area; Grade 2, 25–40% decrease in any height and 20–40% decrease in area; and Grade 3, 40% decrease in any height and area. Vertebral fractures were defined as Grade 1 or higher.

### 2.4. Statistics

The distribution of normality was assessed using the Shapiro–Wilk normality test. Continuous variables are shown as the mean and standard deviation or median and interquartile range, and the groups were compared using the Student’s unpaired t-test. Categorical variables are shown as the number of patients and percentage (%), and the groups were compared using the chi-square test. Univariate logistic regression models were used to examine the factors associated with vertebral fractures, and the results were presented as odds ratio (OR) with 95% confidence interval (CI). Multivariate logistic regression models were created that include variables associated with liver cirrhosis and variables that were statistically significant (*p* < 0.05) in the univariate models. The OR of vertebral fractures was estimated for every SD decrease in BMD and TBS. The C statistic and 95% CI were used to evaluate the discriminative power of each model. The C statistic estimates the area under the receiver operating characteristic curve (AUC) and indicates the model’s ability to discriminate between patients with and without vertebral fractures. AUC was compared using the Hanley–McNeil test [20]. Factors associated with TBS were evaluated using a multivariate linear regression model. Collinearity was assessed using the variance inflation factor, with higher values (≥5) indicating the presence of multicollinearity in a set of multiple regression variables [21]. The correlation between TBS and the variables of interest was analyzed using the Pearson correlation coefficient. The significance threshold was set at *p* < 0.05. All analyses were performed using JMP version 9.0.2 software (SAS Institute Inc., Cary, NC, USA).

## 3. Results

### 3.1. Patients’ Characteristics

Of the 275 patients evaluated, 128 (47%) were women, with a mean age of 72 years, a body mass index (BMI) of 24.0 kg/m^2^, and a MELD score of 9 (Table 1). The distribution of patients in each Child–Pugh class was 73% in A, 21% in B, and 6% in C. Liver cirrhosis was attributed to cryptogenic (33%), hepatitis B virus (19%), hepatitis C virus (16%), alcohol-related liver disease (16%), and other causes (17%). Other causes include primary biliary cholangitis (*n* = 13), autoimmune hepatitis (*n* = 11), and nonalcoholic steatohepatitis (*n* = 22). Women had a higher prevalence of autoimmune hepatitis and primary biliary cholangitis, and a lower prevalence of alcohol-related liver disease than men (*p* < 0.001). In this study, 138 (50%) patients had hepatocellular carcinoma, with a higher prevalence in men than in women (67% vs. 31%; *p* < 0.001).

The mean values of BMD, BMD T-score, and TBS in all enrolled patients were 0.92 g/cm^2^, −0.71, and 1.35, respectively. Of the patients evaluated for bone microarchitecture, 180 (65%) had normal structures (TBS > 1.31), 65 (24%) had partially degraded structures (TBS = 1.23–1.31), and 30 (11%) had degraded structures (TBS < 1.23). Vertebral fractures were noted in 62 (23%) patients, with a higher prevalence in women than in men (33% vs. 14%; *p* < 0.001). Women had lower values related to bone formation and strength than men in terms of 25-hydroxyvitamin D (25[OH]D), BMD, BMD T-score, and TBS. No significant differences between men and women were observed in terms of age, BMI, diabetes mellitus, Child–Pugh class, and MELD score.

### 3.2. Characteristics and Vertebral Fractures

Patients with vertebral fractures were older, had more advanced liver disease (Child–Pugh class B/C), and had lower BMD, BMD T-scores, and TBS than those without vertebral fractures. Men with vertebral fractures were older and had lower TBS than those without vertebral fractures, whereas there were no significant differences in BMD and BMD T-scores between men with and without vertebral fractures (Table 2). Women with vertebral fractures were also older and had lower BMD, BMD T-scores, and TBS than those without vertebral fractures. In each sex, there were no significant differences in BMI, diabetes mellitus, hepatocellular carcinoma, serum levels of calcium and 25(OH)D, and liver function reserves (Child–Pugh class and MELD score) between patients with and without vertebral fractures.

### 3.3. Predictors of Vertebral Fractures

Factors significantly associated with vertebral fractures were age, sex, BMD, and TBS (all *p* < 0.001). The unadjusted OR of vertebral fractures for one standard deviation decrease in TBS and BMD was 2.14 (95% CI, 1.69–2.73; *p* < 0.001) and 1.55 (95% CI, 1.26–1.90; *p* < 0.001; Table 3). Hanley–McNeil test showed that TBS had a significantly higher AUC to discriminate the presence of vertebral fractures than BMD (0.76 vs. 0.69; *p* = 0.040; Appendix A). The OR for TBS in men and women was 3.65 (95% CI, 1.78–7.51; *p* < 0.001) and 2.09 (95% CI, 1.46–3.00; *p* < 0.001). Compared with normal structure, the OR for TBS in partially degraded and degraded structure were 2.30 (95% CI, 1.14–4.64; *p* = 0.020) and 15.17 (95% CI, 6.22–36.98; *p* < 0.001), respectively (Appendix A). Multivariate analysis showed that TBS was significantly associated with vertebral fractures (OR, 2.22; 95% CI, 1.49–3.03; *p* < 0.001; Table 3).

Among the evaluated patients, 41 (15%) were on osteoporosis medications, including bisphosphonate (*n* = 21), selective estrogen receptor modulators (*n* = 1), and 25(OH)D (*n* = 19). However, TBS remained statistically significant (OR, 2.77; 95% CI, 1.68–4.54; *p* < 0.001), even when patients on osteoporosis medications were excluded from the analysis (Appendix A). In the subgroup analyses, the association between TBS and vertebral fractures was observed in the HCC and other etiology groups, but such association was not observed in the liver decompensation group (Appendix A).

### 3.4. Predictors of TBS

Univariate analysis revealed that the factors significantly associated with TBS were age, BMI, MELD score, calcium, 25(OH)D, and BMD. There was no significant association between TBS and HCC (*p* = 0.188), cirrhosis etiology (*p* = 0.832), or liver decompensation (*p* = 0.992). Multivariate analysis showed that age (β = −0.211, *p* < 0.001), BMI (β = −0.251, *p* < 0.001), and BMD (β = 0.583, *p* < 0.001) were significantly associated with TBS (Table 4). The variance inflation factor ranged from 1.06 to 1.29, indicating that there was no multicollinearity in the model. These three factors were significantly associated with TBS in men; however, BMI was not associated with TBS in women. Similar results were obtained when patients on osteoporosis medications were excluded from the analysis (Appendix A). In addition, subgroup analysis also showed that age, BMI, and BMD were significantly associated with TBS, regardless of HCC (Appendix A).

TBS was positively correlated with BMD (r = 0.572; *p* < 0.001; Figure 1a) and negatively correlated with age (r = −0.214; *p* < 0.001; Figure 1b) and BMI (r = −0.137; *p* = 0.023; Figure 1c). The correlation coefficient was more pronounced in women than in men in terms of BMD (r = 0.550 vs. r = 0.492) and age (r = −0.365 vs. r = −0.096), whereas the relationship between TBS and BMI was more pronounced in men (r = −0.284 vs. r = 0.026) (Appendix A).

The correlation coefficient between TBS and BMD is positive (r = 0.572, *p* < 0.001), whereas the correlation is negative for age (r = −0.214, *p* < 0.001) and BMI (r = −0.137, *p* = 0.023). The correlation between TBS and the variables of interest is analyzed using the Pearson correlation coefficient. BMD, bone mineral density; BMI, body mass index; TBS, trabecular bone score.

## 4. Discussion

Secondary osteoporosis is common in cirrhotic patients, regardless of the etiology or severity of liver disease [6]. Because cirrhosis-related osteoporotic fractures are associated with disability, decreased quality of life, and an increased risk of death, all cirrhotic patients should undergo bone densitometry during their clinical course [4]. In addition, recent evidence has shown that osteoporosis can be safely treated with risedronate in patients with liver cirrhosis [22]. BMD is a useful indicator to diagnose osteoporosis [7,8,9]. However, BMD measurement alone is insufficient for screening patients at high risk for osteoporotic fracture because it often occurs in individuals without a reduced BMD [9,10].

The usefulness of TBS in estimating the risk of osteoporotic fractures has been confirmed in patients with secondary osteoporosis, such as long-term corticosteroid use and type 2 diabetes [16]. Individuals with low TBS are at high risk for osteoporotic fractures, and TBS can predict various fractures, independent of BMD, clinical risk factors, and the FRAX tool [15]. However, to the best of our knowledge, there are no reports on the relationship between TBS and vertebral fractures in patients with cirrhosis. Therefore, the results of this study may increase the possibility of an association between TBS and vertebral fractures in patients with cirrhosis. In this study, TBS was lower in patients with vertebral fractures than in those without vertebral fractures. In addition, TBS is helpful in the risk stratification of patients with vertebral fractures, regardless of age, sex, BMD, and Child–Pugh and MELD scores. These findings confirm the usefulness of TBS in cirrhosis-related secondary osteoporosis and provide new insights for a more accurate assessment of osteoporotic fracture risk in patients with cirrhosis.

Low BMD increases the risk of osteoporotic fractures, although it is not the only factor that defines bone strength. Approximately 70% of bone strength can be explained by bone density and the remaining 30% by bone quality, such as bone microarchitecture [23,24,25]. Therefore, only 10–44% of most types of fractures can be directly attributed to osteoporosis (BMD T-score < −2.5) [10], highlighting the importance of the assessment of bone microarchitecture in predicting osteoporotic fractures. Several cross-sectional studies in postmenopausal women have shown that TBS is associated with a higher prevalence of vertebral fractures, with an adjusted OR ranging from 1.97 to 3.81 [26,27,28]. Many prospective studies have also demonstrated that TBS independently predicts the occurrence of vertebral fractures, with adjusted risk ratios ranging from 1.46 to 1.54 [29,30,31]. The findings of this study showed that TBS is associated with vertebral fractures in patients with liver cirrhosis. Therefore, TBS may be useful in screening cirrhotic patients at high risk for vertebral fractures.

BMD is affected by vertebral osteoarthritis and fractures, which leads to an overestimation of BMD measurements and consequently an underestimation of osteoporotic fracture risk [18]. Therefore, TBS may predict vertebral fractures better than BMD [14]. In addition, TBS in the lumbar spine declines earlier than BMD, and TBS may be more sensitive than BMD as a predictor of osteoporotic fractures [32]. Our findings suggest that TBS may also be useful in identifying men with vertebral fractures. Meta-analysis has demonstrated that the ability of TBS to predict osteoporotic fractures is comparable in men and women [15]. Since osteoporosis in men is often overlooked [33,34], the widespread implementation of TBS measurement in clinical practice may contribute to improved fracture risk assessment in all patients, including cirrhotic patients, regardless of sex.

In the present study, there was a positive correlation between TBS and BMD. This finding is consistent with several reports observing that TBS is positively correlated with lumbar spine BMD [29,30,31,35,36]. With aging, the bone remodeling balance becomes negative, resulting in bone loss, disruption of bone microarchitecture, and an increased risk of bone fragility [3]. The negative correlation between TBS and age was more pronounced in women in the present study. This is because the rate of age-related decline in TBS is faster in women than in men [37,38]. For larger individuals with greater soft tissue thickness, DXA detectors recognize images with lower contrast [11]. Since TBS is based on the variation in an individual’s gray level, images with low contrast will underestimate the TBS value. We found that BMI was independently correlated with TBS after adjusting for confounding variables. Meta-analysis studies have also shown a weak negative correlation between TBS and BMI [15]; however, it remains unclear because the calculation of TBS includes soft tissue adjustments estimated from BMI.

This study has several limitations. First, the cross-sectional nature of this study does not allow us to accurately quantify the utility of TBS in the longitudinal prediction of vertebral fractures in patients with cirrhosis. Second, the findings of this study may be subject to unmeasured factors, such as nutritional status, daily physical activity and regular exercise, smoking status, and thyroid, parathyroid, and sex hormones [8]. The patient proportion of cryptogenic cirrhosis was relatively high (92/275; 33.5%), which may lead to a possible patient selection bias. In addition, not all patients underwent histological examination for the diagnosis of cirrhosis. Thus, the results of this study may be affected by selection bias and may not be generalized to other cohorts in different regions and clinical settings. Third, since TBS measurements are optimized for a BMI range of 15–35 kg/m^2^, TBS assessment is not valid for subjects with a BMI above this range [11,12]. However, even when these patients (*n* = 8) were excluded from the analysis, TBS remained significantly associated with vertebral fractures (OR, 1.92; 95% CI: 1.27–2.88). Finally, because the presence of ascites underestimates the actual BMD value by 4.2–7% [6], it is recommended that BMD should be measured immediately after paracentesis. Although this was not performed in this study, TBS remained statistically significant even when patients with ascites (*n* = 58) were excluded from the analysis (OR, 1.89; 95% CI, 1.24–2.88).

## 5. Conclusions

In conclusion, TBS is useful in identifying patients with cirrhosis at risk for vertebral fractures. The present study raises the possibility that TBS has an advantage over BMD in the estimation of vertebral fracture risk in patients with cirrhosis. This study also expands our knowledge of the usefulness of TBS in patients with cirrhosis and illustrates the role of TBS in providing complementary information to BMD in assessing the risk of vertebral fractures. Prospective studies with a larger sample size are warranted to validate our results.

## Figures and Tables

**Figure 1 jcm-11-01562-f001:**
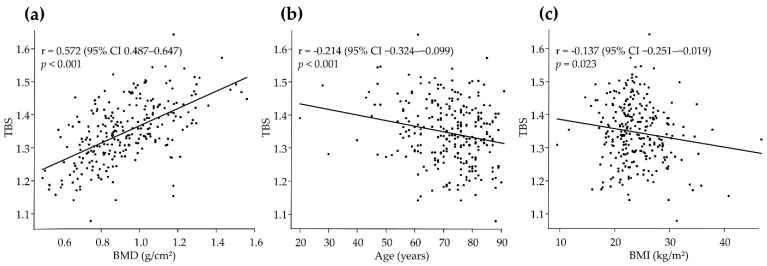
Correlation coefficient between TBS and (**a**) BMD, (**b**) age, and (**c**) BMI.

**Table 1 jcm-11-01562-t001:** Comparison of clinical characteristics between male and female patients.

	Total	Men	Women	*p*-Value ^†^
Characteristic	(*n* = 275)	(*n* = 147)	(*n* = 128)	
Age (years)	72 (12)	72 (11)	71 (13)	0.467
Body mass index (kg/m^2^)	24.0 (4.5)	23.6 (4.0)	24.4 (5.0)	0.115
Diabetes mellitus	83 (30)	49 (33)	34 (27)	0.238
Etiology				
Cryptogenic/HBV/HCV/Alcohol/Others	92/51/43/43/46	48/31/25/34/9	44/20/18/9/37	<0.001
Hepatocellular carcinoma	138 (50)	98 (67)	40 (31)	<0.001
Child–Pugh class				
A/B/C	202/57/16	114/25/8	88/32/8	0.241
MELD score	8 (7–10)	8 (7–9)	8 (7–10)	0.535
Total bilirubin (mg/dL)	1.0 (0.7–1.4)	0.9 (0.6–1.3)	1.1 (0.7–1.5)	0.086
Creatinine (mg/dL)	0.74 (0.62–0.92)	0.84 (0.69–1.03)	0.64 (0.54–0.76)	<0.001
INR	1.04 (0.99–1.15)	1.03 (0.98–1.10)	1.09 (0.99–1.19)	0.060
Albumin (g/dL)	3.8 (3.3–4.1)	3.8 (3.4–4.2)	3.7 (3.3–4.1)	0.431
Sodium (mEq/L)	139 (137–140)	139 (137–140)	139 (138–141)	0.041
Calcium (mg/dL)	9.1 (8.8–9.4)	9.1 (8.8–9.4)	9.2 (8.8–9.5)	0.448
Phosphorus (mg/dL)	3.7 (6.8)	3.2 (0.5)	3.4 (0.5)	0.001
25(OH)D (ng/mL)	13 (10–18)	15 (11–20)	12 (9–15)	<0.001
BMD (g/cm^2^)	0.92 (0.20)	0.99 (0.20)	0.84 (0.17)	<0.001
BMD T-score	−0.71 (1.69)	−0.14 (1.67)	−1.37 (1.47)	<0.001
TBS	1.35 (0.09)	1.37 (0.08)	1.31 (0.09)	<0.001
Vertebral fracture	62 (23)	20 (14)	42 (33)	<0.001

Values are presented as number (percentage), mean (standard deviation), or median (interquartile range). ^†^ Clinical characteristics between the two groups were compared using the chi-square test for categorical variables or the unpaired t-test for continuous variables. BMD, bone mineral density; HBV, hepatitis B virus; HCV, hepatitis C virus; INR, international normalized ratio; MELD, model for end-stage liver disease; 25(OH)D, 25-hydroxyvitamin D; TBS, trabecular bone score.

**Table 2 jcm-11-01562-t002:** Clinical characteristics of the presence or absence of vertebral fractures in the total cohort and in each sex.

	Total (*n* = 275)			Men (*n* = 147)			Women (*n* = 128)		
	No Vertebral Fracture	Vertebral Fracture	*p*-Value ^†^	No Vertebral Fracture	Vertebral Fracture	*p*-Value ^†^	No Vertebral Fracture	Vertebral Fracture	*p*-Value ^†^
Characteristic	(*n* = 213)	(*n* = 62)		(*n* = 127)	(*n* = 20)		(*n* = 86)	(*n* = 42)	
Age (years)	70 (12)	78 (8)	<0.001	71 (11)	78 (8)	0.005	67 (13)	78 (9)	<0.001
Body mass index (kg/m^2^)	23.8 (4.0)	24.6 (5.9)	0.186	23.4 (3.7)	24.4 (5.9)	0.324	24.3 (4.4)	24.7 (6.0)	0.611
Diabetes mellitus	64 (30)	19 (31)	1.000	43 (34)	6 (30)	0.804	21 (24)	13 (31)	0.523
Etiology									
cryptogenic/HBV/HCV/alcohol/others	71/42/28/36/36	9/9/15/7/22	0.245	41/30/20/27/9	7/1/5/7/0	0.160	30/12/8/9/27	14/8/10/0/10	0.046
Hepatocellular carcinoma	106 (50)	32 (52)	0.885	84 (66)	14 (70)	0.804	22 (26)	18 (43)	0.067
Child–Pugh class									
A/B/C	160/38/15	42/19/1	0.045	98/21/8	16/4/0	0.662	62/17/7	26/15/1	0.098
MELD score	8 (7–10)	8 (7–10)	0.673	8 (7–9)	7 (7–10)	0.862	8 (7–10)	8 (7–10)	0.410
Total bilirubin (mg/dL)	1.0 (0.7–1.5)	0.9 (0.5–1.3)	0.107	0.9 (0.7–1.4)	0.7 (0.5–1.1)	0.065	1.1 (0.8–1.5)	1.1 (0.5–1.5)	0.221
Creatinine (mg/dL)	0.73 (0.62–0.91)	0.76 (0.64–0.97)	0.648	0.84 (0.69–0.98)	0.91 (0.80–1.17)	0.305	0.62 (0.54–0.70)	0.72 (0.57–0.80)	0.239
INR	1.06 (0.99–1.17)	1.03 (0.99–1.14)	0.593	1.04 (0.98–1.12)	1.03 (1.00–1.06)	0.607	1.10 (0.99–1.21)	1.04 (0.98–1.15)	0.127
Albumin (g/dL)	3.8 (3.3–4.3)	3.8 (3.4–4.0)	0.923	3.8 (3.3–4.3)	3.8 (3.5–4.1)	0.545	3.7 (3.3–4.3)	3.8 (3.3–4.0)	0.703
Sodium (mEq/L)	139 (137–140)	139 (137–140)	0.965	139 (137–140)	139 (137–140)	0.556	139 (138–141)	139 (138–141)	0.954
Calcium (mg/dL)	9.1 (8.8–9.4)	9.1 (8.9–9.4)	0.778	9.1 (8.8–9.4)	9.0 (8.9–9.3)	0.701	9.2 (8.8–9.5)	9.2 (8.8–9.5)	0.699
Phosphorus (mg/dL)	3.3 (0.5)	3.4 (0.4)	0.017	3.2 (0.5)	3.4 (0.4)	0.059	3.4 (0.5)	3.5 (0.4)	0.539
25(OH)D (ng/mL)	13 (10–18)	12 (9–18)	0.176	15 (11–21)	15 (10–20)	0.774	12 (10–215	11 (9–16)	0.782
BMD (g/cm^2^)	0.95 (0.19)	0.83 (0.20)	<0.001	1.00 (0.20)	0.97 (0.22)	0.567	0.88 (0.17)	0.76 (0.15)	<0.001
BMD T-score	−0.47 (1.63)	−1.53 (1.67)	<0.001	−0.10 (1.64)	−0.35 (1.84)	0.537	−1.02 (1.45)	−2.10 (1.26)	<0.001
TBS	1.37 (0.08)	1.28 (0.09)	<0.001	1.38 (0.08)	1.31 (0.08)	<0.001	1.34 (0.08)	1.27 (0.09)	<0.001

Values are presented as the number (percentage), mean (standard deviation), or median (interquartile range). ^†^ Clinical characteristics between the two groups were compared using the chi-square test for categorical variables or the unpaired t-test for continuous variables. BMD, bone mineral density; HBV, hepatitis B virus; HCV, hepatitis C virus; INR, international normalized ratio; MELD, model for end-stage liver disease; 25(OH)D, 25-hydroxyvitamin D; TBS, trabecular bone score.

**Table 3 jcm-11-01562-t003:** Factors associated with vertebral fractures.

	Univariate		Multivariate	
Characteristics	OR (95% CI)	*p*-Value	OR (95% CI)	*p*-Value
Age	1.10 (1.06–1.14)	<0.001	1.10 (1.06–1.15)	<0.001
Sex (female)	3.10 (1.70–5.64)	<0.001	0.79 (0.31–2.02)	0.616
Child–Pugh score	0.99 (0.83–1.18)	0.887	0.94 (0.70–1.26)	0.682
MELD score	0.98 (0.88–1.08)	0.672	1.02 (0.87–1.21)	0.785
BMD ^†^	1.55 (1.26–1.90)	<0.001	1.04 (0.81–1.33)	0.758
TBS ^†^	2.14 (1.69–2.73)	<0.001	2.22 (1.49–3.03)	<0.001

^†^ Estimated per one standard deviation decrease. AUC, area under the receiver operating characteristic curve; BMD, bone mineral density; CI, confidence interval; MELD, model for end-stage liver disease; OR, odds ratio; TBS, trabecular bone score.

**Table 4 jcm-11-01562-t004:** Factors associated with trabecular bone score ^†^.

Predictors	Partial Regression Coefficient (B)	Standard Error	T-Value	*p*-Value	Standardized Partial Regression Coefficient (β)	VIF
Total						
Age	−0.002	<0.001	−4.43	<0.001	−0.211	1.07
BMI	−0.005	<0.001	−5.31	<0.001	−0.251	1.06
BMD	0.266	0.022	12.20	<0.001	0.583	1.08
Men						
Age	−0.002	<0.001	−3.65	<0.001	−0.237	1.10
BMI	−0.008	0.001	−6.30	<0.001	−0.404	1.07
BMD	0.240	0.026	9.10	<0.001	0.591	1.10
Women						
Age	−0.002	<0.001	−2.49	0.014	−0.204	1.28
BMI	−0.002	0.001	−1.52	0.131	−0.116	1.11
BMD	0.290	0.045	6.44	<0.001	0.529	1.29

^†^ After adjustment for age, BMI, MELD score, calcium, 25(OH)D, and BMD, all of which were significant (*p* < 0.05) in univariate analysis. BMI, body mass index; BMD, bone mineral density; VIF, variance inflation factor.

## Data Availability

The data presented in this study are available upon request from the corresponding author.

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
