# Peer review of "Usefulness of the Trabecular Bone Score in Assessing the Risk of Vertebral Fractures in Patients with Cirrhosis"

_jcm, 2022, doi:10.3390/jcm11061562_

Round 1

Reviewer 1 Report

To the author:

The topic of this manuscript is novel, and the content is clear. Some issues need further modification.

  1. The authors’ name should be checked. “Tatsunori Hanai1” may be wrong.
  2. The information “Citation: Lastname, F.; Lastname, F.; Lastname, F. Title. J. Clin. Med. 2022, 11, x. https://doi.org/10.3390/xxxxx” is confusing.
  3. The authors said “Receiver operating characteristic (ROC) curve analysis was performed to evaluate the discriminative ability of different variables, and the results were presented as the area under the ROC curve (AUC)”. Are you sure that you have done ROC analysis?
  4. In the Table 2 of the section “Characteristics and Vertebral Fractures”, why do not you add the comparison of clinical characteristics in all patients regardless of gender?
  5. The format of table 2 should be improved.
  6. The "association between trabecular bone score and vertebral fractures adjusted for age, sex, and bone mineral density" has been mentioned in the section "Predictors of Vertebral Fractures", but why do not you adjust other factors? Child-Pugh and MELD scores (PMID: 26937922) are important models for evaluation of prognosis of liver cirrhosis. They should be adjusted in these analyses.
  7. The authors said “Multivariate analysis showed that age (β = -0.211, P < 0.001), BMI (β = -0.251, P < 0.001), and BMD (β = 0.583, P < 0.001) were significantly correlated with TBS, with no multicollinearity in the model”. Give the detailed values to demonstrate “no multicollinearity”.
  8. How to improve the osteoporosis should be mentioned a bit in the Discussion. Recently, there is some evidence regarding use of risedronate in liver cirrhosis (PMID: 31831865). Discuss them a bit.
  9. The conclusion mentioned "The present study makes a significant contribution to the research...". However, the grade of current evidence is weak. Additionally, this work did not compare the difference of TBS versus BMD in assessing fracture risk. Generally, your conclusions should be greatly improved.

Author Response

Responses to Reviewer 1

Thank you very much for reviewing our manuscript and offering valuable advice. We appreciate your comments, which have helped us to improve our manuscript. Please find below detailed responses to the reviewer’s comments.

Comments:

  1. The authors’ name should be checked. “Tatsunori Hanai1” may be wrong.

Thank you for pointing this out. The correct name is “Tatsunori Hanai”. We have revised the manuscript accordingly.

  1. The information “Citation: Lastname, F.; Lastname, F.; Lastname, F. Title. J. Clin. Med. 2022, 11, x. https://doi.org/10.3390/xxxxx” is confusing.

We take it that the reviewer is asking about the information provided in the margin of the title page. Since we are instructed to use a Microsoft Word template to prepare our manuscript, we consider that we are not allowed to change this part of the document. We would be grateful if you would accept this explanation.

  1. The authors said “Receiver operating characteristic (ROC) curve analysis was performed to evaluate the discriminative ability of different variables, and the results were presented as the area under the ROC curve (AUC)”. Are you sure that you have done ROC analysis?

Thank you for your useful comment. To improve clarity, we have revised the manuscript as follows: the model’s ability to discriminate between groups was assessed using the area under the receiver operating characteristic curve (AUC) (Page 3, lines 122–123).

  1. In the Table 2 of the section “Characteristics and Vertebral Fractures”, why do not you add the comparison of clinical characteristics in all patients regardless of gender?

Based on the reviewer's comments, we have performed further analysis to compare the clinical characteristics of patients with and without vertebral fractures in all patients. The results showed that patients with vertebral fractures were older, had more advanced liver disease (Child-Pugh class B/C), and had lower BMD, BMD T-scores, and TBS than those without vertebral fractures. We have added this information in the revised manuscript (Page 4, lines 158–160; and revised Tables 2). Thank you for the useful suggestion.

  1. The format of table 2 should be improved.

In line with the reviewer’s comment, we have improved the format of Table 2 to make it easier to understand.

  1. The "association between trabecular bone score and vertebral fractures adjusted for age, sex, and bone mineral density" has been mentioned in the section "Predictors of Vertebral Fractures", but why do not you adjust other factors? Child-Pugh and MELD scores (PMID: 26937922) are important models for evaluation of prognosis of liver cirrhosis. They should be adjusted in these analyses.

We agree that the association between TBS and vertebral fractures needs to be adjusted for Child-Pugh and MELD scores. After adjusting for the aforementioned factors, Child-Pugh score, and MELD score, TBS was still significantly associated with vertebral fractures (OR, 2.26; 95% CI, 1.52–3.34; P < 0.001), with AUC of 0.84 (95% CI, 0.78–0.89). We have added this information in the manuscript (Page 5, lines 186–188; Page 7, lines 246–247). Thank you for the valuable suggestion.

  1. The authors said “Multivariate analysis showed that age (β = -0.211, P < 0.001), BMI (β = -0.251, P < 0.001), and BMD (β = 0.583, P < 0.001) were significantly correlated with TBS, with no multicollinearity in the model”. Give the detailed values to demonstrate “no multicollinearity”.

The reviewer has raised concerns regarding supporting data to demonstrate no multicollinearity. As described in the Statistics section, collinearity was assessed using the variance inflation factor. The high value (variance inflation factor ≥ 5) indicates the presence of multicollinearity in a set of multiple regression variables (Page 3, lines 124–126). Table 4 shows that the variance inflation factor ranged from 1.06 to 1.29, indicating that there was no multicollinearity in the model (Page 6, lines 205–206). Thank you for the useful comment.

  1. How to improve the osteoporosis should be mentioned a bit in the Discussion. Recently, there is some evidence regarding use of risedronate in liver cirrhosis (PMID: 31831865). Discuss them a bit.

The reviewer has requested that we provide information concerning the treatment of osteoporosis in patients with liver cirrhosis. As the reviewer noted, recent evidence has showed that osteoporosis can be safely treated with risedronate in patients with liver cirrhosis. We have added this information along with the relevant reference in the revised manuscript (Page 7, lines 232–233; and new Reference #20). We thank the reviewer’s valuable suggestion.

  1. Lima, T.B.; Santos, L.A.A.; Nunes, H.R.C.; Silva, G.F.; Caramori, C.A.; Qi, X.; Romeiro, F.G. Safety and efficacy of risedronate for patients with esophageal varices and liver cirrhosis: a non-randomized clinical trial. Sci Rep 2019, 9, 18958, doi:10.1038/s41598-019-55603-y.

  1. The conclusion mentioned "The present study makes a significant contribution to the research...". However, the grade of current evidence is weak. Additionally, this work did not compare the difference of TBS versus BMD in assessing fracture risk. Generally, your conclusions should be greatly improved.

Thank you for the useful comments. To address the point raised by the reviewer, we carried out further data analysis to compare the differences between TBS and BMD in fracture risk assessment. Hanley-McNeil test showed that TBS had a significantly higher AUC to discriminate the presence of vertebral fractures than BMD (0.76 vs. 0.69; P = 0.040). We also agree with the reviewer’s point regarding the grade of current evidence and toned down the language in the Conclusion section as follows: the present study raises the possibility that TBS has an advantage over BMD in the estimation of vertebral fracture risk in patients with cirrhosis. We have added this description in the revised manuscript (Page 5, lines 178–180; and Page 8, lines 303–304).

In closing, let me thank you once again for your comments which have helped us to improve the quality of our paper. We hope that the above responses meet with the approval of the editors and reviewers.

Reviewer 2 Report

This is a cross-sectional study aimed to determine the predictive validity of trabecular bone score (TBS) for vertebral fractures in cirrhotic patients. Some concerns in the methodology have been raised.

Major concerns:

  1. Study novelty: Although the usefulness studies of TBS remain lacking in cirrhotic patients, the usefulness of TBS in estimating the risk of osteoporotic fractures has been confirmed in patients with secondary osteoporosis.    
  2. Study design: Was this study a retrospective or prospective study? How to deal with the missing data?
  3. The causal relationship should not be overexplained by an association study, such as TBS “predicts” vertebral fracture with an OR of 2.26.
  4. Histological features were listed as a method in the diagnosis of liver cirrhosis in the manuscript; however, it is unclear if all the study subjects received a histological examination for the diagnosis of liver cirrhosis. The clinical characteristics of liver cirrhosis may be presented in the later stage of cirrhotic patients. The possible selection bias coming from the diagnosis criteria should be addressed.
  5. Confounding factors: Around 50% patients suffered from hepatocellular carcinoma (HCC). However, it is unclear if HCC itself or its treatments may be related to vertebral fractures (the outcome). The associations of TBS with other clinical factors, such as HCC, other etiology, and liver decompensation (Child-Pugh class B/C), should be examined. In addition, some sensitivity tests for these patient subgroups may be helpful for further clarification.
  6. The “others” etiology of liver cirrhosis is unclear and should be explained.
  7. The AUC data of TBS should be compared to those of bone marrow density (BMD).

Author Response

Responses to Reviewer 2

Thank you very much for reviewing our manuscript and offering valuable advice. We appreciate your comments, which have helped us to improve our manuscript. Please find below detailed responses to the reviewer’s comments.

Major comments:

  1. Study novelty: Although the usefulness studies of TBS remain lacking in cirrhotic patients, the usefulness of TBS in estimating the risk of osteoporotic fractures has been confirmed in patients with secondary osteoporosis.

We really understand that the usefulness of TBS in estimating the risk of vertebral fractures has already been confirmed in patients with secondary osteoporosis. The present study might be insufficient to provide novel insights into the investigation of bone disease in patient at risk for osteoporosis and vertebral fractures. However, to our knowledge, there is little evidence on the usefulness of TBS in estimating the risk of osteoporotic fractures in cirrhotic patients. Considering that osteoporosis is often overlooked in cirrhotic patients and that they are approximately twice as likely to develop osteoporotic fractures as non-cirrhotic patients, we believe that the results of this study will be helpful to people working on measures to improve osteoporosis in these patients. We would be grateful if you would accept this explanation.

  1. Study design: Was this study a retrospective or prospective study? How to deal with the missing data?

Thank you for your comment. This is a cross-sectional study and there is no missing data.

  1. The causal relationship should not be overexplained by an association study, such as TBS “predicts” vertebral fracture with an OR of 2.26.

We agree that the cross-sectional study design does not allow us to conclude the causal relationship. Based on the reviewer’s comment, we have revised the sentence as follows: the findings of this study showed that TBS is associated with vertebral fractures with an OR of 2.26 after adjustment for age, sex, and BMD in patients with liver cirrhosis (Page 7, line 260 to Page 8, line 261). We appreciate your useful comment.

  1. Histological features were listed as a method in the diagnosis of liver cirrhosis in the manuscript; however, it is unclear if all the study subjects received a histological examination for the diagnosis of liver cirrhosis. The clinical characteristics of liver cirrhosis may be presented in the later stage of cirrhotic patients. The possible selection bias coming from the diagnosis criteria should be addressed.

The reviewer is concerned about the possible selection bias coming from the diagnosis criteria for liver cirrhosis. In the present study, not all patients underwent histological examination for the diagnosis of liver cirrhosis. Liver cirrhosis was diagnosed by each hepatologist in our hospital based on a combination of clinical features, laboratory variables, medical imaging features, and, if possible, histological features. We really understand that the results of this study may be affected by selection bias. We have added this information to the limitation section of the revised manuscript (Page 2, line 82; and Page 8, lines 290–292).

  1. Confounding factors: Around 50% patients suffered from hepatocellular carcinoma (HCC). However, it is unclear if HCC itself or its treatments may be related to vertebral fractures (the outcome). The associations of TBS with other clinical factors, such as HCC, other etiology, and liver decompensation (Child-Pugh class B/C), should be examined. In addition, some sensitivity tests for these patient subgroups may be helpful for further clarification.

The reviewer has requested that we provide information concerning the associations of TBS with other clinical factors, such as HCC, other etiology, and liver decompensation (Child-Pugh class B/C). Based on the reviewer's comments, we have performed further analysis and found that there was no significant association between TBS and HCC (P = 0.188), cirrhosis etiology (P = 0.832), or liver decompensation (P = 0.992). Subgroup analysis also showed that age, BMI, and BMD were significantly correlated with TBS, regardless of HCC. We have added this information in the revised manuscript (Page 6, lines 201–203; Page 6, lines 209–210; and new Supplemental Table 4). Thank you for the valuable suggestion.

  1. The “others” etiology of liver cirrhosis is unclear and should be explained.

Thank you for your comments. Other causes include primary biliary cholangitis (n = 13), primary biliary cholangitis (11), nonalcoholic steatohepatitis (n = 22), and cryptogenic cirrhosis (n = 92). We have added this information in the revised manuscript (Page 3, lines 136–138).

  1. The AUC data of TBS should be compared to those of bone marrow density (BMD).

To address the point raised by the reviewer, we carried out further data analysis to compare the differences between TBS and BMD in fracture risk assessment. Hanley-McNeil test showed that TBS had a significantly higher AUC to discriminate the presence of vertebral fractures than BMD (0.76 vs. 0.69; P = 0.040). We have added this description in the revised manuscript (Page 5, lines 178–180). Thank you for the useful suggestion.

In closing, let me thank you once again for your comments which have helped us to improve the quality of our paper. We hope that the above responses meet with the approval of the editors and reviewers.

Reviewer 3 Report

This manuscript shows usefulness of TBS score in assessing the risk of vertebral fractures in patients with cirrhosis. The research design is excellent. The description is clear. Daily physical activity of patients is of great interest because it seems to have a significant impact on bone microarchitecture, but the authors explicity mention in the research limitations as an unmeasured factors.

P3 line 104: Dose it mean that "vertebral compression fracture" already exists?

Author Response

Responses to Reviewer 3

Thank you very much for reviewing our manuscript and offering valuable advice. We appreciate your comments, which have helped us to improve our manuscript. Please find below detailed responses to the reviewer’s comments.

Comment:

P3 line 104: Dose it mean that "vertebral compression fracture" already exists?

Thank you for your comment. It means that pre-existing vertebral compression fractures do not affect the measurement of TBS. Based on the reviewer’s comment, we have revised the sentence (Page 3, lines 104).

In closing, let me thank you once again for your comments which have helped us to improve the quality of our paper. We hope that the above responses meet with the approval of the editors and reviewers.

Round 2

Reviewer 1 Report

I am familiar with the ROC analysis of one variable. However, I am not sure about how to perform the ROC analysis after adjusting for confounding factors. In the sentence "After adjusting for the aforementioned factors, Child-Pugh score, and MELD score, TBS was still significantly associated with vertebral fractures (OR, 2.26; 95% CI, 1.52–3.34; P < 0.001), with AUC of 0.84 (95% CI, 0.78–0.89)", how did you do a adjusted ROC analysis? Give more information in the Statistics section.

Give more information about how to compare the two AUCs?

Give a reference for the sentence "Collinearity was assessed using the variance inflation factor, with higher values (≥ 5) indicating the presence of multicollinearity in a set of multiple regression variables".

Author Response

Responses to Reviewer 1

Thank you very much for reviewing our manuscript and offering valuable advice. We appreciate your comments, which have helped us to improve our manuscript. Please find below detailed responses to the reviewer’s comments.

  1. I am familiar with the ROC analysis of one variable. However, I am not sure about how to perform the ROC analysis after adjusting for confounding factors. In the sentence "After adjusting for the aforementioned factors, Child-Pugh score, and MELD score, TBS was still significantly associated with vertebral fractures (OR, 2.26; 95% CI, 1.52–3.34; P < 0.001), with AUC of 0.84 (95% CI, 0.78–0.89)", how did you do a adjusted ROC analysis? Give more information in the Statistics section.

We apologize for the inadequate explanation in the Statistics section. Based on the reviewer’s comment, we have revised the relevant description as follows: Univariate logistic regression models were used to examine factors associated with vertebral fractures, and the results were presented as odds ratio (OR) with 95% confidence interval (CI). Multivariate logistic regression models were created including variables of interest and variables that were statistically significant (P < 0.05) in the univariate models. The OR of vertebral fractures was estimated for every SD decrease in BMD and TBS. The C statistic and 95% CI were used to evaluate the discriminative power of each model. The C statistic estimates the area under the receiver operating characteristic curve (AUC) and indicates the model's ability to discriminate between patients with and without vertebral fractures (Page 3, lines 121–129). We hope this answers the points raised by the reviewer. Thank you for the suggestive comment.

  1. Give more information about how to compare the two AUCs?

AUC was compared using the Hanley-McNeil test. We have added this information along with the relevant reference in the revised manuscript (Page 3, lines 129–130; and new Reference #20). We thank the reviewer’s valuable suggestion.

  1. Hanley, J.A.; McNeil, B.J. A method of comparing the areas under receiver operating characteristic curves derived from the same cases. Radiology 1983, 148, 839-843, doi:10.1148/radiology.148.3.6878708.

  1. Give a reference for the sentence "Collinearity was assessed using the variance inflation factor, with higher values (≥ 5) indicating the presence of multicollinearity in a set of multiple regression variables".

We have cited a relevant reference in the text (new Reference #21). Thank you for the advice.

  1. Jame, G.; Witten, D.; Hastie, T.; Tibshirani, R. An Introduction to Statistical Learning: with Applications in R, 2nd ed.; Springer: Berlin, Germany, 2021; pp. 102.

In closing, let me thank you once again for your comments which have helped us to improve the quality of our paper. We hope that the above responses meet with the approval of the editors and reviewers.

Reviewer 2 Report

This is a cross-sectional study aimed to determine the predictive validity of trabecular bone score (TBS) for vertebral fractures in cirrhotic patients. Some concerns in the methodology have been raised.

Major concerns:

  1. Study design: Was this cross-sectional study conducted in a retrospective or prospective manner? How many cirrhotic patients were not included in this study? The possibility of sampling error should be addressed.
  2. The causal relationship should not be overexplained by this association study throughout the whole manuscript. Please carefully revise the related statements in the Discussion.
  3. The authors misunderstood the subgroup sensitivity analysis for hepatocellular carcinoma (HCC), other etiology, and liver decompensation. As the data shown in Table 3, the associations between TBS and vertebral fracture should be further examined in the subgroups of HCC, other etiology, and liver decompensation.
  4. The most common cause in “others” etiology of liver cirrhosis was cryptogenic (92/138; 66.7%) in this study; therefore, most patients (92/275; 33.5%) suffered from cryptogenic cirrhosis. The patient proportion of cryptogenic cirrhosis was even higher than those of patients with viral hepatitis. In Japan, the most common cause of liver cirrhosis should be HCV infection. The possible patient selection bias should be considered.
  5. The AUC data of TBS should be compared to those of bone marrow density (BMD): As the data shown in Table 3, the AUC data of BMD should be clearly presented and compared in a Table.

Author Response

Responses to Reviewer 2

Thank you very much for reviewing our manuscript and offering valuable advice. We appreciate your comments, which have helped us to improve our manuscript. Please find below detailed responses to the reviewer’s comments.

Major concerns:

  1. Study design: Was this cross-sectional study conducted in a retrospective or prospective manner? How many cirrhotic patients were not included in this study? The possibility of sampling error should be addressed.

Thank you for your comment. This cross-sectional study was conducted in a prospective manner, and 24 patients with liver cirrhosis who were not evaluated for BMD, TBS, or vertebral fractures were excluded from the data analysis. As pointed out by the reviewer, we are very aware of the possibility of sampling error. To address this concern, we hope to be able to survey a larger number of participants in the future so that we can reduce the possibility of sampling error. We have added this information in the revised manuscript (Page 2, lines 73–76). We appreciate your useful comment.

  1. The causal relationship should not be overexplained by this association study throughout the whole manuscript. Please carefully revise the related statements in the Discussion.

Based on the reviewer’s comment, we have revised the relevant statements in the Discussion to avoid overexplaining the causal relationship (Page 7, line 247; Page 7, line 264 to Page, 8 line 265; and Page 8, lines 271–272). We thank the reviewer’s appropriate suggestion.

  1. The authors misunderstood the subgroup sensitivity analysis for hepatocellular carcinoma (HCC), other etiology, and liver decompensation. As the data shown in Table 3, the associations between TBS and vertebral fracture should be further examined in the subgroups of HCC, other etiology, and liver decompensation.

We apologize for the misunderstanding regarding the subgroup sensitivity analysis. Based on the reviewer’s comments, we have conducted further analyses to clarify the associations between TBS and vertebral fracture in the subgroups of HCC, other etiology, and liver decompensation. Subgroup analyses showed that the associations between TBS and vertebral fractures was observed in the subgroups of HCC and other etiology, but not in liver decompensation. We have added this information in the revised manuscript (Page 5, lines 195–197; and new Supplemental Table 4).

  1. The most common cause in “others” etiology of liver cirrhosis was cryptogenic (92/138; 66.7%) in this study; therefore, most patients (92/275; 33.5%) suffered from cryptogenic cirrhosis. The patient proportion of cryptogenic cirrhosis was even higher than those of patients with viral hepatitis. In Japan, the most common cause of liver cirrhosis should be HCV infection. The possible patient selection bias should be considered.

Thank you for pointing this out. We really understand that the this is critical limitation of this study. The present study was conducted in the tertiary hospital, and this may result in the relative high proportion of cryptogenic cirrhosis. Because of the possible patient selection bias, the results of this study may not be generalized to other cohorts in different regions and clinical settings. We have added this description in the limitation section (Page 8, lines 294–299).

  1. The AUC data of TBS should be compared to those of bone marrow density (BMD): As the data shown in Table 3, the AUC data of BMD should be clearly presented and compared in a Table.

Based on the reviewer’s comment, we have provided the information on the AUC data of BMD in the form of a Table (new Supplemental Table 1). Thank you for the useful suggestion.

In closing, let me thank you once again for your comments which have helped us to improve the quality of our paper. We hope that the above responses meet with the approval of the editors and reviewers.

This manuscript is a resubmission of an earlier submission. The following is a list of the peer review reports and author responses from that submission.